# Model-Informed Target Morning 17α-Hydroxyprogesterone Concentrations in Dried Blood Spots for Pediatric Congenital Adrenal Hyperplasia Patients

**DOI:** 10.3390/ph16030464

**Published:** 2023-03-21

**Authors:** Viktoria Stachanow, Uta Neumann, Oliver Blankenstein, Nele Alder-Baerens, Davide Bindellini, Peter Hindmarsh, Richard J. Ross, Martin J. Whitaker, Johanna Melin, Wilhelm Huisinga, Robin Michelet, Charlotte Kloft

**Affiliations:** 1Department of Clinical Pharmacy and Biochemistry, Institute of Pharmacy, Freie Universitaet Berlin, Kelchstr 31, 12169 Berlin, Germany; viktoria.stachanow@fu-berlin.de (V.S.); davide.bindellini@fu-berlin.de (D.B.); charlotte.kloft@fu-berlin.de (C.K.); 2Graduate Research Training Program, PharMetrX, 12169 Berlin, Germany; 3Charité-Universitätsmedizin, Freie Universität Berlin, 13353 Berlin, Germany; uta.neumann@charite.de (U.N.);; 4Labor Berlin, Charité Vivantes GmbH, 13353 Berlin, Germany; nele.alder-baerens@charite.de; 5Developmental Endocrinology Research Group, UCL Institute of Child Health, London WC1E 6BT, UK; 6Department of Oncology and Metabolism, University of Sheffield, Sheffield S10 2TN, UK; r.j.ross@sheffield.ac.uk (R.J.R.); martin.whitaker@sheffield.ac.uk (M.J.W.); 7Institute of Mathematics, Universität Potsdam, 14476 Potsdam, Germany; huisinga@uni-potsdam.de

**Keywords:** congenital adrenal hyperplasia, 17α-hydroxyprogesterone, dried blood spots, target concentration range, pediatrics, pharmacometrics

## Abstract

Monitoring cortisol replacement therapy in congenital adrenal hyperplasia (CAH) patients is vital to avoid serious adverse events such as adrenal crises due to cortisol underexposure or metabolic consequences due to cortisol overexposure. The less invasive dried blood spot (DBS) sampling is an advantageous alternative to traditional plasma sampling, especially in pediatric patients. However, target concentrations for important disease biomarkers such as 17α-hydroxyprogesterone (17-OHP) are unknown using DBS. Therefore, a modeling and simulation framework, including a pharmacokinetic/pharmacodynamic model linking plasma cortisol concentrations to DBS 17-OHP concentrations, was used to derive a target morning DBS 17-OHP concentration range of 2–8 nmol/L in pediatric CAH patients. Since either capillary or venous DBS sampling is becoming more common in the clinics, the clinical applicability of this work was shown by demonstrating the comparability of capillary and venous cortisol and 17-OHP concentrations collected by DBS sampling, using a Bland-Altman and Passing-Bablok analysis. The derived target morning DBS 17-OHP concentration range is a first step towards providing improved therapy monitoring using DBS sampling and adjusting hydrocortisone (synthetic cortisol) dosing in children with CAH. In the future, this framework can be used to assess further research questions, e.g., target replacement ranges for the entire day.

## 1. Introduction

Congenital adrenal hyperplasia (CAH) is a form of adrenal insufficiency most commonly leading to a lack of the enzyme P450c21 and, therefore, to cortisol deficiency. The accumulation of cortisol precursors, including dehydroepiandrosterone and androstenedione (adrenal androgens), often results in clinical signs of virilization or hirsutism in female patients and acceleration of skeletal age in both sexes [1,2,3]. Cortisol deficiency in CAH requires life-long hormone replacement therapy with hydrocortisone (synthetic cortisol) in a daily dose of 10–15 mg/m^2^ given three times per day to the growing patient [4]. Cortisol replacement needs to be carefully monitored to avoid adverse events such as adrenal crisis (cortisol underexposure) or Cushing’s syndrome (cortisol overexposure) [2,4].

Dried blood spot (DBS) sampling, when compared to plasma blood sampling, is a less invasive option to monitor cortisol and further steroid marker concentrations. DBS sampling only requires full blood volumes of approximately 20 µL, obtained via a finger prick and dropped on filter paper [5,6,7]. The feasibility of DBS sampling, as well as long-term stability at room temperature, was investigated and confirmed for CAH-relevant steroids in DBS in previous literature [8,9,10]. Therefore, DBS sampling can facilitate therapeutic monitoring for pediatric CAH patients. However, therapeutic monitoring using DBS measurements remains a challenge since no target blood biomarker concentrations are currently established for the DBS methodology. Analyte concentrations collected by DBS relate linearly to plasma or serum concentrations, differing by a factor specific to the analyte [7] or nonlinearly, as is observed for cortisol [11]. 

Knowing target morning biomarker concentrations is vital for successful CAH therapy monitoring. A key biomarker for guiding replacement therapy in CAH is 17α-hydroxyprogesterone (17-OHP), a precursor of cortisol and adrenal androgens, which accumulates in CAH patients. The administration of hydrocortisone downregulates adrenocorticotropic hormone (ACTH) and sequentially the 17-OHP concentration by negative feedback [1,2,12]. The target morning 17-OHP concentration range in traditionally measured plasma has been suggested to be 12–36 nmol/L (dependent on body surface area) in CAH patients [2] but is still unknown for concentrations collected by DBS sampling. The aim of this analysis was to derive a DBS 17-OHP morning target concentration range using nonlinear mixed effects (NLME) modeling, a pharmacometric modeling and simulation approach. In contrast to classical regression methods, NLME modeling enables the quantification of the relationship between, e.g., plasma and DBS concentrations by considering all available data simultaneously, thereby quantifying within- and between-subject-variability and explaining this variability using subject-specific characteristics [13]. The modeling and simulation-derived target range can pave the way for guidance for clinicians when investigating biomarker concentrations using DBS sampling and could have the potential to improve therapy monitoring in pediatric CAH patients.

## 2. Methods

To derive a target morning 17-OHP concentration range using DBS sampling, a modeling and simulation framework was developed, leveraging different sources of drug pharmacokinetics (PK, i.e., cortisol concentrations) and biomarker pharmacodynamics (PD, i.e., 17-OHP concentrations), and exploring their relationships. This framework approach was chosen because of the limited availability of data from clinical studies or routine clinical data collection, i.e., clinical study data including plasma cortisol concentrations in pediatric CAH patients, venous 17-OHP concentrations from simultaneous DBS measurements, endogenous plasma cortisol concentrations in healthy children and 17-OHP target morning concentrations only known in plasma (Figure 1). An overview of the population characteristics in all used datasets can be found in the Appendix A.

The following sections describe the different steps of this framework in detail, namely: (A) the development of a PK/PD model linking plasma cortisol concentrations and venous DBS 17-OHP concentrations based on a pediatric CAH patient study, (B) a Bland-Altman and Passing-Bablok regression analysis to investigate the comparability of simultaneously measured venous and capillary 17-OHP concentrations, because in clinical practice both matrices can be used for DBS sampling in pediatric CAH patients; and finally, (C1) the derivation of a DBS 17-OHP target morning range by leveraging the developed PK/PD model and plasma cortisol data of non-CAH children and (C2 and C3) assessing the plausibility of the derived 17-OHP target morning concentration from DBS sampling by comparing to an expected target range, based on 17-OHP target morning concentration range in plasma, known from the literature.

### 2.1. Step A: Establishing a Quantitative Relationship between Plasma Cortisol (PK) and Venous DBS 17-OHP (PD) Concentrations in Pediatric CAH Patients: PK/PD Clinical Trial Data and Model Development

#### 2.1.1. Data and Graphical Evaluation

To determine a relationship between pediatric plasma cortisol and venous DBS 17-OHP concentrations, data from a phase 3 trial for the pediatric hydrocortisone formulation Alkindi^®^ (Cardiff Medicentre, Cardiff, UK) (ClinicalTrials.gov Identifier: NCT02720952) [14,15] was used (Figure 1, Box A). In this study, 24 pediatric adrenal insufficiency patients (23 CAH patients, 1 hypopituitarism patient) received an individualized morning dose of Alkindi^®^ (Cardiff Medicentre, Cardiff, UK). For the PK/PD model development, the data of 12 young children, aged 2–6 years and of 6 infants, aged 28 days-2 years old, was used. Total plasma cortisol concentrations were available prior to dosing, 1 h and 4 h post-dose and in the young children cohort at 2 additional time points. Further details on the clinical trial data are described in Michelet et al. [16]. Additionally, venous DBS cortisol and 17-OHP concentrations were measured in the same patients [11]. The data used for PK/PD modeling were graphically analyzed, investigating the DBS 17-OHP concentrations as a function of time and of plasma cortisol concentrations (R (4.0.2) and R Studio (1.3.1056) [17,18]). 

#### 2.1.2. PK/PD Modeling

For the development of the PK/PD model characterizing the relationship between cortisol in plasma and 17-OHP in venous DBS, NLME modeling was applied [13]. The model development was based on a previously published pediatric hydrocortisone PK model [11,16]. A sequential approach was used [19] in which the individual PK model parameters from this previously published PK model [11] were included in the model dataset, and the PD parameters were estimated.

Since 22 (25%) out of 88 venous DBS 17-OHP concentrations were below the lower limit of quantification (LLOQ, <1.3 nmol/L: 22.5% in young children and 35.3% in infants), the so-called M3 method [20] was applied (for more details see Appendix A).

For the PK/PD model development, the software packages NONMEM (7.4.3, ICON, Dublin, Ireland) and Perl speaks NONMEM 4.7.0 (Uppsala University, Uppsala, Sweden) were used, embedded in the workbench Pirana (version 2.9.6) [21,22]. Model evaluation and selection were performed using standard diagnostics such as the objective function value (OFV), parameter plausibility, relative standard errors (RSE), goodness-of-fit (GOF) plots and visual predictive checks (VPC, *n* = 1000). 

### 2.2. Step B: Comparison of Capillary and Venous DBS Cortisol and 17-OHP Concentrations from Routine Monitoring in Pediatric CAH Patients: Bland-Altman and Passing-Bablok Regression Analysis 

In clinical practice, either venous or capillary DBS samples are usually obtained in pediatric CAH patients. However, the trial data used for the PK model [11] and PK/PD model development in step (A) included, besides plasma cortisol concentrations, venous (not capillary) DBS cortisol and 17-OHP concentrations only. To assess the comparability between capillary and venous DBS cortisol and 17-OHP concentrations, a Bland–Altman analysis as well as a Passing–Bablok regression were conducted (Figure 1, Box B) [23,24]. CAH monitoring data obtained from routine clinical sampling at the Charité-University Hospital Berlin was used for the analysis, including DBS cortisol and 17-OHP concentrations from venous and capillary blood, simultaneously measured in 15 pediatric patients aged from 2 months to 11 years (median: 8 years). One parallel venous and capillary DBS sample was obtained from each patient between the morning and late afternoon and between 1–9 h after the last HC administration. 

In the Bland-Altman analysis, the differences in the measurements in the two matrices were plotted versus the mean of the capillary and venous concentrations. Due to the large concentration range, the relative method differences were presented. The agreement interval was defined as within the mean difference ± 1.96*standard deviation (SD) [23,25].

For the Passing-Bablok regression, scatter plots of venous versus capillary DBS concentrations were generated, including a regression line, a 95% confidence interval (CI) of its intercept and slope, as well as a regression function. The intercept and slope of the line of identity laying within the CI of the regression were used to demonstrate whether there was no constant and proportional difference between the two examined methods [24,26]. 

### 2.3. Step C: Derivation and Evaluation of Venous and Capillary DBS 17-OHP Target Morning Concentration Range for Pediatric CAH Patients: PK/PD Model Predictions

The developed PK/PD model (Figure 1, Box A) was applied to derive a DBS 17-OHP target morning concentration range in simulations (*n* = 1000; Figure 1, Box C1). To predict physiological (=healthy) DBS 17-OHP concentrations, physiological plasma cortisol concentrations, densely sampled over 24 h from 28 non-CAH children aged from 5 to 9 years (Figure 2) [27,28], were used as the PK (plasma cortisol) input, inhibiting the synthesis of 17-OHP. The 17-OHP compartment was initialized with the median 17-OHP baseline, i.e., morning concentrations observed in the young children and infants in the Alkindi^®^ (Cardiff Medicentre, Cardiff, UK) trial (=14.4 nmol/L). The prediction focused on physiological DBS 17-OHP concentrations simulated between 6 and 8 a.m. in order to cover the monitoring time range of interest, corresponding to the time before the morning dose. 

To avoid a higher risk for adverse events, it is advised not to let the 17-OHP concentrations decrease to physiological ranges in CAH therapy [1]. Indeed, among clinical experts, the target 17-OHP concentrations are assumed to be approximately 3–5 times higher than physiological 17-OHP concentrations. Thus, the simulated physiological DBS 17-OHP morning concentrations were multiplied by 3 and by 5 to approximate the DBS 17-OHP target morning concentration range.

To assess the plausibility of the simulation-derived DBS 17-OHP target morning concentration range, a simple calculation of an expected DBS 17-OHP target morning range was performed (Figure 1, Box C2). The calculation of the expected range was based on baseline 17-OHP concentrations which were simultaneously measured in plasma and in DBS in the same young children and infants who were investigated during the Alkindi^®^ (Cardiff Medicentre, Cardiff, UK) study. The ratio of these plasma and DBS 17-OHP baseline concentrations observed in the two age groups as well as the (adult) plasma 17-OHP target morning concentration range known from the literature [2], were used for determining the expected target morning range to which the model-based derived target morning range was then compared (Figure 1, Box C3). 

## 3. Results

### 3.1. Step A: Establishing a Quantitative Relationship between Plasma Cortisol (PK) and Venous DBS 17-OHP (PD) Concentrations in Pediatric CAH Patients: PK/PD Clinical Trial Data and Model Development

#### 3.1.1. Data and Graphical Evaluation

The observed concentration-time profiles of DBS 17-OHP (Figure 3A) indicated a trend towards a u-shape relationship, which was more pronounced in the young children patient cohort due to the longer sampling interval. This u-shape trend indicated an initial decrease of 17-OHP, likely due to the hydrocortisone treatment-mediated inhibition of 17-OHP synthesis, followed by a subsequent increase of 17-OHP due to decreasing plasma cortisol concentrations after hydrocortisone administration. The result of these mechanisms/processes was also visible in DBS 17-OHP versus plasma cortisol concentrations (Figure 3B), revealing 17-OHP decreasing with increasing cortisol concentrations, except the highest cortisol concentrations measured right after hydrocortisone administration, i.e., before the feedback mechanism set in (gray box in Figure 3B). The DBS 17-OHP concentrations, including the concentrations at baseline, had similar ranges in young children and infants.

#### 3.1.2. PK/PD Modeling

Based on the exploratory graphical analysis, the PK/PD model was developed to describe a cortisol-mediated inhibition of the 17-OHP synthesis, concretely on its synthesis rate k_syn_ (indirect response model; Figure 4). The full PK/PD model scheme (Appendix A), information on the estimated pediatric individual parameters from Stachanow et al. [11], which were part of the dataset, corresponding model equations and the NONMEM model code can be found in Appendix A.

The maximum inhibitory effect (I_max_) was pre-defined as 1 (=100% inhibition), and the Hill Factor (Hill) was fixed to 1 after obtaining estimates close to 1 (Table 1). The synthesis rate constant k_syn_ was defined as the product of the estimated first-order degradation rate constant (k_deg_) and the DBS 17-OHP concentration at baseline (17-OHP_BASE_). The cortisol concentration inhibiting 50% of the maximum inhibitory effect (IC_50_) was estimated to be 21 nmol/L with a high estimated interindividual variability (IIV on IC_50_ = 104 %CV), which was not explained by any covariate present in the dataset, such as body weight, corticosteroid-binding globulin (CBG) or albumin. Relative standard errors of the estimated model parameters were low, except for IIV on IC_50_ (53%).

The GOF plots and the VPC indicated that the venous DBS 17-OHP observations were appropriately described by the PK/PD model (Appendix A).

### 3.2. Step B: Comparison of Capillary and Venous DBS Cortisol and 17-OHP Concentrations from Routine Monitoring in Pediatric CAH Patients: Bland-Altman and Passing-Bablok Regression Analysis

To investigate whether venous DBS concentrations (as in step (A)) are comparable to capillary DBS concentrations, which can be obtained in clinical practice as well, the Bland-Altman and Passing-Bablok regression analyses were conducted (Figure 1, Box B). The Bland-Altman analysis (Figure 5A,B) did not show any substantial bias between capillary and venous DBS measurements for the drug (cortisol mean difference: +3.13%) and 17-OHP (mean difference: +3.73%). Almost all data points except one each (93% of both cortisol and 17-OHP measurements) were within the range of the mean difference ± 1.96*SD. Moreover, the Passing–Bablok regression also showed an agreement between the venous and capillary DBS concentrations since, for both cortisol and 17-OHP, the lines of identity lay within the CIs of the regression lines (Figure 5C,D). The slopes of the regression lines for cortisol and 17-OHP were close to 1 (1.0 and 0.7, respectively), and the y-intercept for cortisol was close to 0 (0.8), indicating a high similarity between capillary and venous DBS concentrations.

Thus, the capillary and venous DBS concentrations were considered comparable for cortisol and 17-OHP. This finding allowed us to use the developed PK/PD model for deriving a 17-OHP target morning range for DBS sampling, which is applicable for both venous and capillary DBS 17-OHP concentrations (Figure 1, Box C1). 

### 3.3. Step C: Derivation and Evaluation of Venous and Capillary DBS 17-OHP Target Morning Concentration Range for Pediatric CAH Patients: PK/PD Model Predictions

The PK/PD model-based derivation of the DBS 17-OHP morning target concentration range for pediatric CAH patients resulted in a range of 2.1–8.3 nmol/L (Figure 6, median: 4.4 nmol/L) after multiplying simulated physiological (=healthy) concentrations by factors 3 and 5 as assumed ratios for the target-to-physiological 17-OHP concentrations (see Section 2.3). The derived target range (interquartile range) excluded 25% of the lowest and 25% of the highest concentrations of all simulated physiological DBS 17-OHP morning (6–8 a.m.) concentrations due to the very high variability in the prediction. 

To assess the plausibility of the derived target range (Figure 1, box C3), an approximately expected DBS 17-OHP target morning range was calculated (Figure 1, box C2) based on the known (adult) 17-OHP target morning concentration range in plasma (12–36 nmol/L) [2], divided by the median plasma/DBS 17-OHP morning concentration ratio of 9.29 which was observed in young children and infants (*n* = 18) during the Alkindi^®^ (Cardiff Medicentre, Cardiff, UK) trial (Appendix A). 

Since the DBS 17-OHP target morning range (2.1–8.3 nmol/L), derived from the modeling and simulation framework analysis, was in the same order of magnitude as the approximately expected range (1.3–3.9 nmol/L), indicated by the red horizontal lines (Figure 6), the results were judged as plausible.

## 4. Discussion

We developed a modeling and simulation framework based on data from different matrices (i.e., plasma, venous and capillary DBS samples) and different sources (i.e., clinical trials, clinical routine and literature data) due to the limited availability of data. The framework included a PK/PD model which successfully linked DBS 17-OHP to plasma cortisol concentrations in pediatric CAH patients. By leveraging this framework, we were able to derive a plausible target morning concentration range for the clinically important biomarker 17-OHP, for DBS sampling in pediatric CAH patients, in the range of 2–8 nmol/L, which is applicable for both venous or capillary DBS samples. 

The PK/PD model captured the known mechanism of administered hydrocortisone inhibiting 17-OHP biosynthesis via the suppression of the HPA-axis by a negative feedback mechanism [1,2,12]. This mechanism was implemented in an indirect response model with 17-OHP synthesis inhibition, well characterizing the u-shape trend observed in the 17-OHP concentration-time profiles. A main limitation of in silico approaches such as PK/PD modeling is that the purpose and validity of the model depend on the data it is based on. The development of the PK/PD model was based on sparse pediatric clinical trial data and therefore did not allow for a more complex model structure, e.g., including covariate relationships. Despite the limited available data, as typical for trials in this vulnerable patient population, the model resulted in plausible parameter estimates with satisfactory precision. The estimated IC_50_ of 21.0 nmol/L (=plasma cortisol concentration leading to 50% of maximum inhibitory effect Imax on 17-OHP synthesis) was in the same order of magnitude as previously determined IC_50_ values, e.g., 40.3 nmol/L by Al-Kofahi et al. [29] and 48.6 nmol/L by Melin et al. [30]. The current lower estimate of IC_50_ can be explained by the difference in matrices between the former studies and our analysis. As in the current study, plasma cortisol PK was linked to PD in full blood obtained via DBS, where DBS cortisol concentrations are substantially lower than plasma concentrations, an approximately 2-fold lower IC_50_ value, in line with reported plasma/DBS cortisol ratio ranges [11], was expected. 

For the derivation of a DBS 17-OHP target morning concentration range, published physiological plasma cortisol concentrations from healthy children aged between 5 and 9 were applied [27,28]. This age range was also covered in the young children cohort in the Alkindi^®^ (Cardiff Medicentre, Cardiff, UK) trial (2–6 years). Besides the young children data, infant data (28 days-2 years) was also included due to the very similar 17-OHP concentrations between these two age groups in the current study. To support this inclusion, we also developed a PK/PD model based on young children only. No significant differences were found in the estimated model parameters, their imprecisions, and the derived DBS 17-OHP concentrations using this further reduced model dataset. The presented model for infants and young children can serve as a basis to obtain DBS 17-OHP target concentrations in, e.g., neonates or older children in the future when corresponding data becomes available. 

Bland-Altman and Passing-Bablok regression analyses were conducted to translate the modeling and simulation results which are based on venous DBS 17-OHP concentrations to clinical routine where, besides venous, also capillary DBS 17-OHP concentrations can be sampled. The analyses demonstrated that capillary and venous DBS measurements of cortisol and 17-OHP are comparable. Thus, the results from our previously published PK model [11], as well as from this PK/PD analysis (both based on venous DBS data), can be applied to capillary DBS samples. 

The plausibility of the derived DBS 17-OHP morning concentration range was assessed by comparing the results to an expected concentration range which was based on the 17-OHP target morning concentration range in plasma, reported in Merke et al. [2]. This expected target range is to be viewed as a simple plausibility check only since the underlying plasma 17-OHP target range applies to adults. To the best of our knowledge, no further 17-OHP target concentrations have been reported. 

Whereas a target range is established for plasma 17-OHP concentrations [2], there are no cortisol target concentrations suggested except for mimicking the physiological circadian rhythm [1]. 17-OHP is a commonly used biomarker as it is a precursor of cortisol and androgens, of which elevated concentrations are closely linked to the clinical signs of CAH. DBS sampling can facilitate regular and less invasive measurement of cortisol and numerous biomarkers [5]. Therefore, besides in plasma, it is also needed to identify biomarker target concentrations in DBS. 

Despite its advantages and high potential, especially for pediatric patients, DBS sampling is not fully established yet in drug and biomarker monitoring. The techniques used for DBS sampling and measurements can vary between laboratories [6,31,32], and therefore further research, the establishment of standardization of DBS sampling and of the corresponding bioanalysis is needed. Furthermore, taking the patients’ hematocrit values into consideration is vital to ensure accurate quantification of the analyte, as the hematocrit influences the spreading of the sampled blood on the DBS filter paper [5,7,33]. Since hematocrit values are known to be higher in neonates [33], these are potentially valuable covariates to be investigated in future analyses, once available. Target biomarker concentrations for alternative child-friendly sampling techniques, such as mouth swabs to measure steroids in saliva, are to be investigated in the future, for which the developed modeling and simulation framework could serve as a starting point.

Since the cortisol-mediated inhibition of the 17-OHP synthesis, which is incorporated into the PK/PD model, does not suffice to characterize the circadian rhythm of 17-OHP during the day, we focused on the most relevant target concentrations in the morning only. Circadian rhythms of 17-OHP and other CAH-relevant analytes, e.g., corticosteroid-binding globulin, have already been quantified within PK model analyses [30,34]. In the future, the presented PK/PD model can be expanded with the impact of ACTH on the circadian rhythm of 17-OHP to suggest target DBS 17-OHP concentration-time profiles indicating the target ranges for any time of the day.

## 5. Conclusions

Within the modeling and simulation framework described here, we obtained a plausible 17-OHP concentrations target range to be measured before the morning hydrocortisone dose for the first time using DBS sampling methodology. This derived target morning range has the potential to develop further and could provide guidance for monitoring young children suffering from CAH in the future.

Further model development is suggested to derive circadian DBS-derived target 17-OHP concentration-time profiles over 24 h to contribute meaningful target concentrations which clinicians can refer to for the treatment of CAH in children.

## Figures and Tables

**Figure 1 pharmaceuticals-16-00464-f001:**
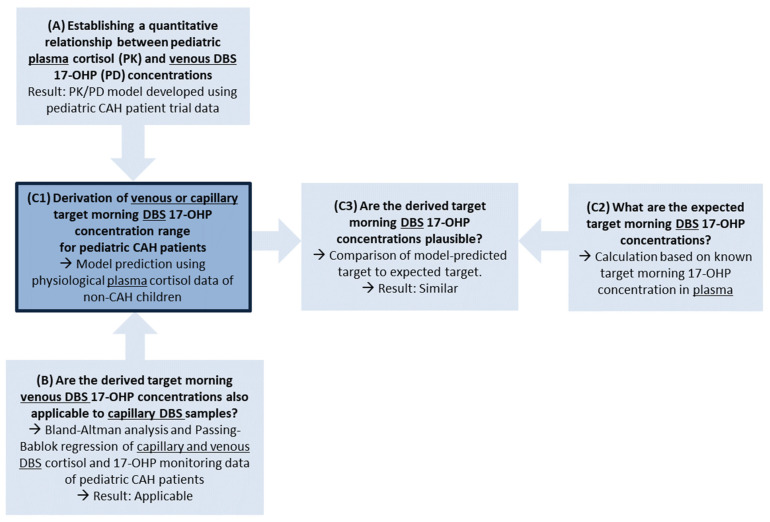
Modeling and simulation framework to derive target morning concentration range for the biomarker 17α-hydroxyprogesterone (17-OHP) sampled from dried blood spots (DBS) in pediatric congenital adrenal hyperplasia (CAH) patients in clinical routine.For the goal of this analysis, the derivation of the target range (dark blue box C1), a pharmacokinetic/pharmacodynamic (PK/PD) model, quantitatively linking plasma cortisol and venous DBS 17-OHP, was developed (A). The applicability of the derived target range, which was based on venous DBS data, to capillary DBS sampling was shown in a Bland–Altman analysis and Passing–Bablok regression (B). To check for plausibility (C3), the derived target range was compared to a calculated expected DBS 17-OHP target range (C2).

**Figure 2 pharmaceuticals-16-00464-f002:**
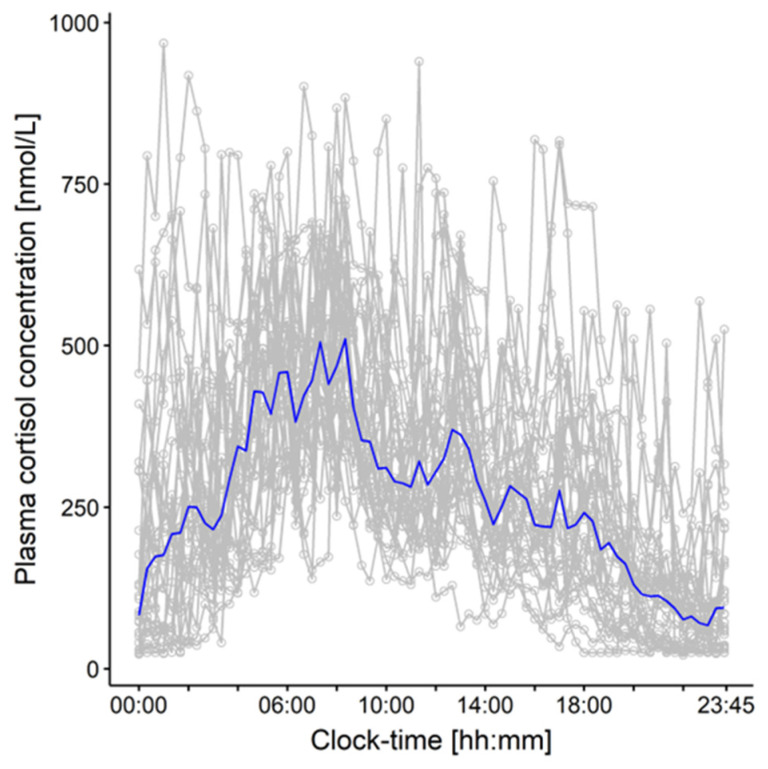
Individual plasma cortisol concentration-time profiles (gray) and median plasma cortisol concentration-time profiles (blue) obtained over 24 h from 28 healthy children aged 5–9 years.

**Figure 3 pharmaceuticals-16-00464-f003:**
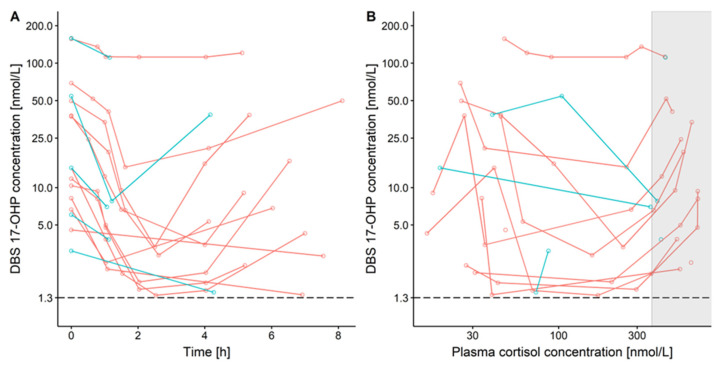
17α-hydroxyprogesterone (17-OHP) concentrations measured in venous dried blood spots (DBS) over time after first (baseline) observation, semi-log scale (**A**) and versus plasma cortisol concentration, log scales (**B**), in pediatric CAH patients receiving hydrocortisone single morning dose. Red: young children, blue: infants, dashed line: lower limit of quantification = 1.3 nmol/L, gray box: highest plasma cortisol concentrations directly after hydrocortisone administration.

**Figure 4 pharmaceuticals-16-00464-f004:**
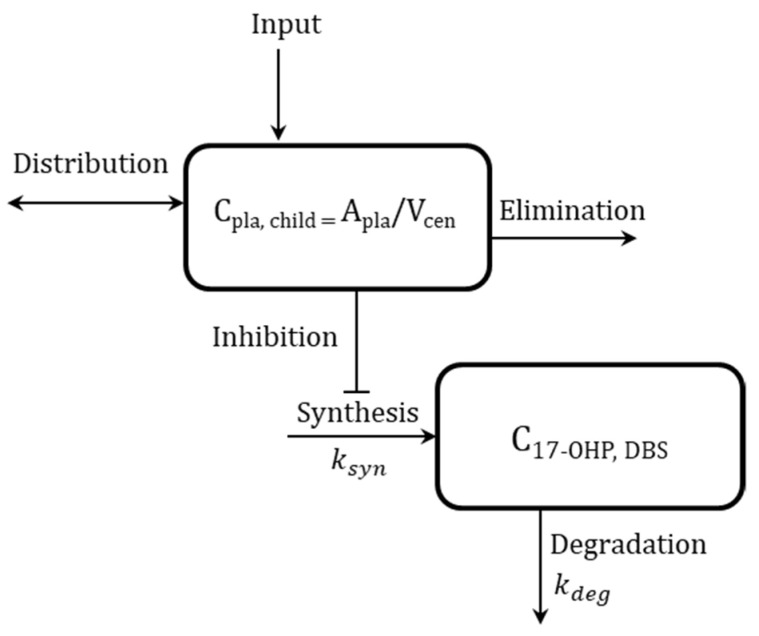
Simplified schematic representation of developed pharmacokinetic/pharmacodynamic (PK/PD) model for cortisol (top box)-mediated inhibition of synthesis of 17α-hydroxyprogesterone (17-OHP; bottom box). Pediatric cortisol concentration in plasma (C_pla, child_), pediatric cortisol amount (A_pla_), central volume of distribution (V_cen_), 17-OHP concentration in dried blood spots (C_17-OHP, DBS_), synthesis rate constant of 17-OHP (k_syn_), and first-order degradation rate constant of 17-OHP (k_deg_). For the full model scheme of the developed PK/PD model, see Appendix A.

**Figure 5 pharmaceuticals-16-00464-f005:**
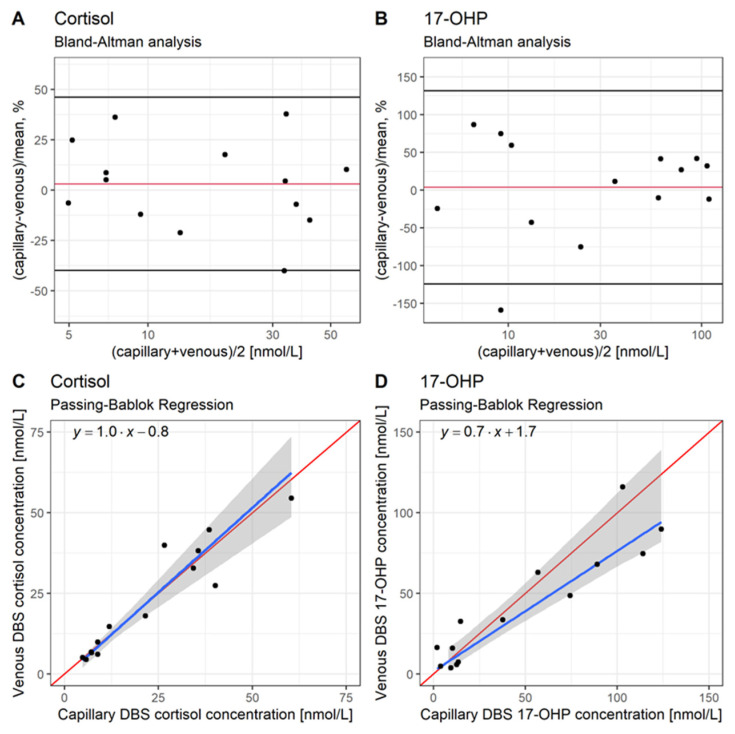
Comparison between capillary and venous cortisol (**A**,**C**) and 17α-hydroxyprogesterone (17-OHP, **A**,**D**) dried blood spot (DBS) concentrations, obtained from 15 pediatric congenital adrenal hyperplasia (CAH) patients. (**A**,**B**): Bland–Altman analysis. Capillary-venous/mean of difference [%] versus the mean of capillary venous cortisol and DBS 17-OHP concentrations. Red line: Mean difference [%], black lines: Mean difference—1.96*SD (standard deviation) and mean difference + 1.96*SD [%]. (**C**,**D**): Passing–Bablok regression. Red line: line of identity, blue line: regression line, gray area: 95% confidence interval for the regression line.

**Figure 6 pharmaceuticals-16-00464-f006:**
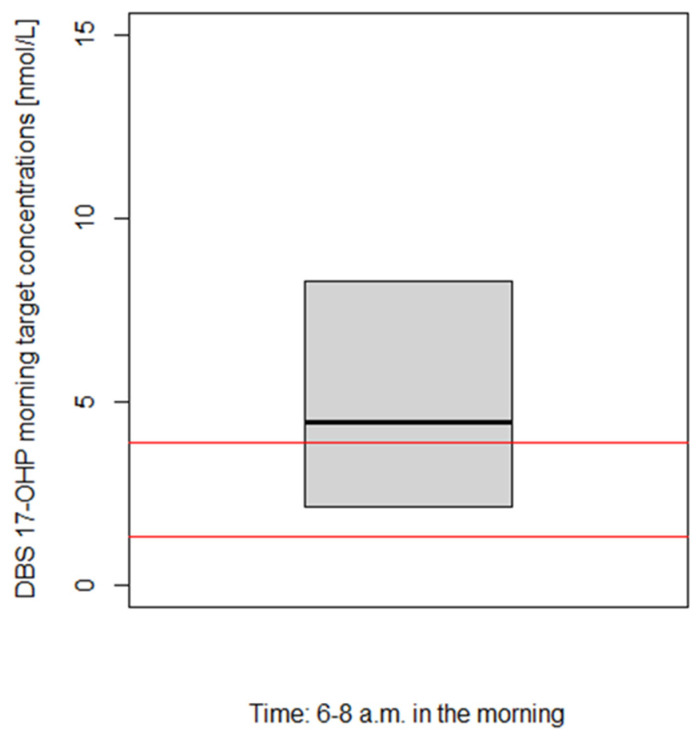
Derived dried blood spot (DBS) 17α-hydroxyprogesterone (17-OHP) target morning concentration range (=model-predicted physiological range multiplied by 3 to 5). Interquartile range (gray box) and median (black line) of the derived DBS 17-OHP target morning concentrations (see step (C1) in text). Red lines: Calculated expected target DBS 17-OHP concentration range in the morning (see step (C2) in text).

**Table 1 pharmaceuticals-16-00464-t001:** Parameter estimates of developed pharmacokinetic/pharmacodynamic (PK/PD) model for cortisol (drug) and 17α-hydroxyprogesterone (biomarker) concentrations in young children and infants.

Parameter	Estimate (RSE, %)
**Structural model**	
k_deg_ [1/h]	1.22 (7.0)
IC_50_ [nmol/L] I_max_ [-] Hill [-]	21.0 (27) 1 * 1 *
**Interindividual variability (IIV)**	
IIV on k_deg_, %CV	5.0*
IIV on IC_50_, %CV	104 (53)
IIV on 17-OHP_BASE_, %CV	131 *
**Residual unexplained variability (RUV)**	
RUV [%CV]	38.1 (15)

* Fixed parameter. Residual variability was estimated by an additive model on a logarithmic scale. 17-OHP_BASE_: 17α-hydroxyprogesterone (17-OHP) dried blood spot concentration at baseline, Hill: Hill coefficient, IC_50_: Cortisol concentration inhibiting 50% of the maximum inhibitory effect I_max_, k_deg_: first-order degradation rate constant of 17-OHP, RSE: Relative standard error.

## Data Availability

Data is contained within the article and Appendix A.

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
