# Peer review of "Model-Informed Target Morning 17α-Hydroxyprogesterone Concentrations in Dried Blood Spots for Pediatric Congenital Adrenal Hyperplasia Patients"

_pharmaceuticals, 2023, doi:10.3390/ph16030464_

Round 1
Reviewer 1 Report
I have gone through the manuscript. It is well structured and written in a systematic way. The article was focused on morning 17-α-hydroxyprogesterone concentrations in dried blood spots for pediatric congenital adrenal hy-3 hyperplasia patients which is very crucial to avoid adrenal crisis. This is very important clinical pharmacy area. I recommend for publication of the paper.
Author Response
Please see document

Reviewer 2 Report
In this study, target morning of biomarker, 17-α-hydroxyprogesterone concentrations in dried blood spots (DBS) was evaluated to monitor cortisol replacement therapy. Modeling and simulation framework, including a pharmacokinetic/pharmacodynamic model linking plasma cortisol concentrations to DBS 17-OHP concentrations, was used to derive a target morning DBS 17-OHP concentration. Overall the manuscript has written well and authors claim novelty as this is first evaluation of analyte in DBS samples. Following comments need to be addressed before its consideration in pharmaceuticals.
1. Introduction section need to be elaborate by addition of previous available method and advantages of this evaluation over previous reported literature.
2. It’s looking that the study is based on in silico modelling and no in vitro and in vivo study was performed, so authors should mention the limitation about application of this method.
3. Some grammatical and syntax error has been observed in text section of the manuscript which need to be polished.
4. Author need to mentioned that how the difference in haematocrit value and blood spot volume can be optimize during evaluation of this method.
5. The stability of the target biomarker in DBS also need to described for routine application of this method.
Author Response
Please see document

Reviewer 3 Report
The authors conducted a study to derive a DBS 17-OHP morning target concentration range using a pharmacometrics modeling and simulation approach. Computer modeling and simulation are important parts of modern research methodology. The work is well written. Though it contains sufficient novelty to be accepted for publication I suggest the following changes.
1. The article's title should make it clear that the study is about modeling and simulation.
2. I suggest preparing a table showing the characteristics of the study cohort.
3. I suggest a subsection: Statistical methodology.
Author Response
Please see document
